# Towards Disentangling Information Paths with Coded ResNeXt

**Apostolos Avranas**[*]
EURECOM
Sophia Antipolis, France
`avranas@eurecom.fr`

**Marios Kountouris**
EURECOM
Sophia Antipolis, France
`kountour@eurecom.fr`

## Abstract

The conventional, widely used treatment of deep learning models as black boxes provides limited or no insights into the mechanisms that guide neural network decisions. Significant research effort has been dedicated to building interpretable models to address this issue. Most efforts either focus on the high-level features associated with the last layers, or attempt to interpret the output of a single layer. In this paper, we take a novel approach to enhance the transparency of the function of the whole network. We propose a neural network architecture for classification, in which the information that is relevant to each class flows through specific paths. These paths are designed in advance before training leveraging coding theory and without depending on the semantic similarities between classes. A key property is that each path can be used as an autonomous single-purpose model. This enables us to obtain, without any additional training and for any class, a lightweight binary classifier that has at least $60\%$ fewer parameters than the original network. Furthermore, our coding theory based approach allows the neural network to make early predictions at intermediate layers during inference, without requiring its full evaluation. Remarkably, the proposed architecture provides all the aforementioned properties while improving the overall accuracy. We demonstrate these properties on a slightly modified ResNeXt model tested on CIFAR-10/100 and ImageNet-1k.

## 1 Introduction

Most successful deep learning architectures for image classification consist of a certain building block applied sequentially several times: one block follows another until a linear operation finally outputs the model prediction. In deep convolutional neural networks (CNNs), the block consists of multiple convolutional operations [36, 37] applied sequentially. Nonetheless, there are numerous proposals placing the convolutional layers in parallel, forming multi-branch designs. For instance, inception models [72, 73] use blocks with multiple branches, each applying some convolutional operations on the block's input and finally concatenating at the end the output of all branches. The multi-branch design framework can also accommodate skip connections [22], as initially done in ResNeXt networks [84], and later refined using squeeze-excitation in [25], or a split-attention mechanism in [88]. A first question we ask in this work is: *what is the purpose of multi-branch architectures?*

Initially, in AlexNet [33] two branches were employed to allow the distribution of the model across two GPUs, which at that time had limited memory. Nowadays, multi-branch architectures are commonly used for distributing the parameters of a block into branches such that each one applies a separate transformation to the input. However, rare are the cases where each branch is shown to contribute in a different way. One example is SKNet [40], in which each branch is associated with different receptive field size, and zooming in or out of an input image activates the appropriate branch.

---

[*]Currently working at Amadeus, Nice

36th Conference on Neural Information Processing Systems (NeurIPS 2022).

Nonetheless, the value of multi-branch networks is mostly justified by achieving a higher accuracy. Multi-branch blocks are also used for network architecture search, where the block/cell architecture is optimized selecting the number of branches, the operation that each performs, and how they are combined [42, 55, 57, 90, 93]. Still, the focus therein was on improving accuracy.

In this work, we investigate how to ensure that each branch provably contributes in a different way in a multi-branch architecture. We propose a novel way to organize in a class-wise manner the transformations carried out by the branches. Before the training starts and without using any information on the semantic properties of the classes, we assign each branch to a specific set of classes. This set remains fixed throughout the training and the branches are trained to activate only to classes within that set. This behavior is achieved mainly by applying a loss function that pushes the output of the branches to be zero for the samples that do not belong to their assigned set of classes. Thanks to this assignment, once the network is trained, for any given class there is a unique path traversing the network through which the information related to that class flows. Conditioning on a class then, extracting only the parameters that participate in its unique path results in a model that has $60\%$ less parameters than the original one and operates as a binary classifier for that class. To showcase the unique features and the advantages of our idea, we use the state-of-the-art multi-branch architecture ResNeXt [84], to which we perform a small number of modifications.

Our main contributions can be summarized as follows:

- We modify the ResNeXt architecture so that it functions in a more transparent way by forcing the information related to each class to flow through well-defined network paths.
- As a proof of concept, we show that in order to form those paths, it is not necessary to rely on the semantics of each class and the similarity between classes.
- Without any additional training, we can obtain a single-purpose model per class operating as a binary classifier, which has $60\%$ fewer parameters as compared to the complete network.
- We demonstrate that the intermediate layers can be used both for making early predictions and for providing a confidence level for correctness of the network's final prediction.
- The proposed Coded ResNeXt significantly improves ResNeXt accuracy across all tested datasets.

## 2 Related Work

There have been numerous attempts to understand how deep neural networks actually work. For instance, activation maximization tries to find the input that increases neuron activation [13, 53, 51, 85]. Saliency maps [67, 65, 68, 70, 54, 58] find the pixels that have the largest influence on the model prediction. However, such approaches cannot really explain how the network operates, and they mainly serve as post hoc visualization methods [59]. In contrast, there are many interpretable by design proposals[78, 89, 6, 49, 31]; however, most of them focus on enhancing interpretability only in the last layers. In [78], the final linear layer is replaced with a differentiable decision tree, and in [89], a loss is used to make each filter of the very high-level convolutional layer represent a specific object part. In [6], the model's output is compared with learnt prototypes, whereas in [31] represents concepts on which humans can intervene.

Another direction for enhancing interpretability is through disentanglement. While there is not yet a generally accepted definition, disentanglement aims at separating the main factors that are present in the data distribution [4, 23, 44]. Still, existing approaches focus on disentangling the factors at a single vector/tensor (the latent representation), which is either the input or the output of a network. This applies to (variational) autoencoders [24, 5, 7, 30, 14], generative adversarial networks [8, 46, 28], normalizing flows [15, 63], or even architectures that aim to decompose the content to the style representation of an image [19, 17]. In contrast, we approach disentanglement as "the way information travels through the network" [60] and look at the neural network architectures for classification as a whole. Specifically, our goal is to control the paths through which information flows by assigning each part of the network to a specific subsets of classes. Related to our work is [80], where they interpret a deep neural network by identifying such information paths; one key difference is that they use a post hoc method, so the paths are identified and not designed.

There are two additional lines of work related to ours. The first line proposes to dynamically control the path that a sample follows through the network by letting an extra network choose which parts to be pruned/omitted [39, 9, 2, 18, 41, 79, 76, 83, 43]. This brings memory and speedup gains during inference, since the samples do not pass through the entire network. Our proposal has a few key

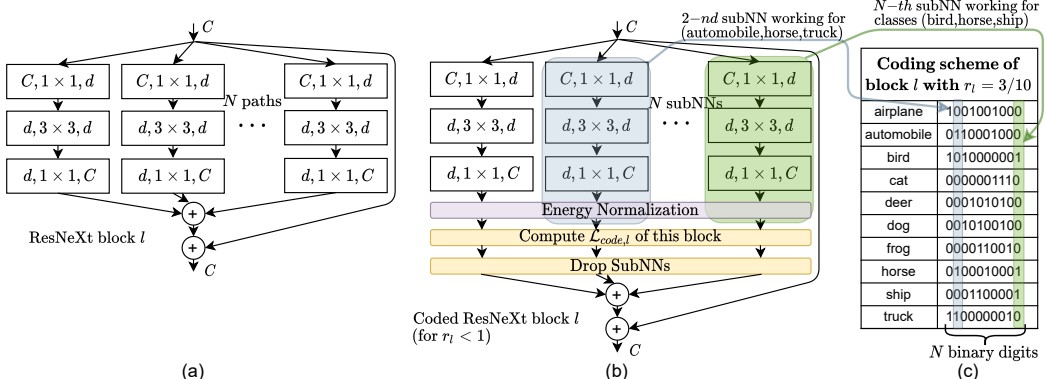

Figure 1: Building block of ResNeXt and the proposed variant. **(a)**: ResNeXt block. A layer is shown as (# in channels, kernel size, # out channels). **(b)**: Coded ResNeXt block. With light violet color we depict the architectural addition and with beige the algorithmic ones. The energy normalization keeps the total sum of the subNNs' output energies constant. Each subNN's output can be zeroed by the dropSubNNs operation with probability $p_{drop}$. Depending on the class of the sample, the loss $\mathcal{L}_{code,l}$ pushes the total energy to be allocated among subNNs with a specific order. **(c)**: An example of the order we used for CIFAR-10. We name this table as the coding scheme of the Coded ResNeXt block; this is defined before training and is kept fixed afterwards. To design the coding scheme we follow some general rules described in Section 3.2.1. The ratio $r_l = 3/10$ means that for the $l$-th block $N_{act,l} = 3$ out of $N = 10$ subNNs will work for each class. The $\mathcal{L}_{code,l}$ tries to match the output energy of the subNNs to their corresponding digit, depending on the binary codeword of the class.

differences with these works. First, we do not use any extra network that has to be trained to learn good paths. Second, our paths are defined prior to training (and thus not learnt). Most importantly, these works do no guarantee that two samples of the same class follow the same path, so it is not possible to extract class-specific network portions to use as single-purpose models. The second line of work concerns models that can make early predictions without evaluating the whole network [64]. Similarly to some previous works [38, 66, 29, 92, 82], we also apply loss functions to the middle layers which facilitates early predictions; however, all those works necessitate additional parameters that are trained as classifiers performing early predictions. In our case, since we force each class to have a unique activation footprint on every middle layer, the early predictions emerge naturally by just looking at the activation patterns created in the middle layers as the samples are forwarded.

While in this work we assign classes to parts of the network arbitrarily, an interesting possibility is to exploit semantic/visual similarities for the assignment, similarly to [11, 12]. This may lead to improvements in performance and interpretability, but it comes with some caveats. In particular, it is not always available or straightforward to obtain these semantic relationships. In [11], for instance, to perform classification on ImageNet, the authors had to resort to another database (WordNet [16]).

## 3 Coded ResNeXt

### 3.1 The block

The typical ResNeXt block [84] is depicted in Fig. 1(a). It takes input $x \in \mathbb{R}^{C \times H \times W}$ ($C$ is the number of input channels and $H$, $W$ are the height and width of the input planes, respectively) and outputs $y$ of the same dimensions. It consists of $N$ paths/branches ($N$ is called cardinality in [84]). Each branch, which is called *sub-neural network* (subNN) here, performs transformation $\mathcal{T}_n$, $n \in \{1, \cdots N\}$ which are all aggregated together with the input $x$, giving the block's output $y$:

$$y = x + \sum_{n=1}^{N} \mathcal{T}_n(x). \tag{1}$$

### 3.1.1 Energy Normalization

For the Coded ResNeXt block depicted in Fig. 1(b), the sole architectural change we introduce is the *Energy Normalization* applied before aggregating the transformed inputs $\mathcal{T}_n(x) \in \mathbb{R}^{C \times H \times W}$. For convenience, let $t_n := \mathcal{T}_n(x)$. If $(t_n)_{c,h,w} \in \mathbb{R}$ is the element of $t_n$ in position $(c, h, w)$, then we define function $\mathcal{E}$ as:

$$\mathcal{E}(t_n) = \frac{1}{CHW} \sum_{c=1}^{C} \sum_{h=1}^{H} \sum_{w=1}^{W} \left((t_n)_{c,h,w}\right)^2, \tag{2}$$

which gives the mean energy of the output signal of the $n$-th subNN. Energy Normalization simply divides the outputs of all branches by a scalar value equal to the square root of the total mean energy, i.e,

$$\bar{t}_n = \frac{t_n}{\sqrt{\mathcal{E}_{avg}}} \text{ with } \mathcal{E}_{avg} = \frac{1}{N} \sum_{i=1}^{N} \mathcal{E}(t_i), \forall n \in \{1, \cdots, N\}. \tag{3}$$

Given that $\mathcal{E}(ax) = a^2 \mathcal{E}(x)$ for scalar $a \in \mathbb{R}_{\geq 0}$, it is easy to see that this step normalizes the total energy, since after it, the sum of the energy of all subNNs becomes $\sum_{n=1}^{N} \mathcal{E}(\bar{t}_n) = N$.

### 3.1.2 Coding Loss

We present here our first algorithmic addition. After the Energy Normalization, we compute a novel loss function, coined *coding loss* $\mathcal{L}_{code}$. Consider a classification problem of $K$ classes. Let $l$ be the index of the position of a ResNeXt block within the network. As seen in Fig. 1(c), for that block, we assign to each class a binary codeword $w_{l,k}, k \in \{1, \cdots, K\}$ of length $N$, indicating which subNNs we want to activate for that class. If the $n$-th subNN operates for class $k$, then the $n$-th digit of $w_{l,k}$ is $(w_{l,k})_n = 1$, and $(w_{l,k})_n = 0$ otherwise. To ensure that each class receives the same number $N_{act,l}$ of operating subNNs, all $K$ codewords are designed with exactly $N_{act,l}$ ones. We define the ratio

$$r_l = \frac{N_{act,l}}{N}, \tag{4}$$

which measures how much each class utilizes the block's total computational resources. We term the mapping of the classes to codewords, as in Fig. 1(c), the *coding scheme* of the block.

Given an input of class $k$, the coding loss forces the mean energies of the subNNs that are inactive for class $k$ to zero and those of the active subNNs to positive values. The coding loss for the $l$-th block is

$$\mathcal{L}_{code,l} = \frac{1}{N} \sum_{n=1}^{N} (r_l \mathcal{E}(\bar{t}_n) - (w_{l,k})_n)^4. \tag{5}$$

Note that after the Energy Normalization, the total subNNs mean energy is $\sum_{n=1}^{N} \mathcal{E}(\bar{t}_n) = N$, while the codeword has $N_{act,l} = r_l N$ ones, hence we multiply $\mathcal{E}(\bar{t}_l)$ by $r_l$.

We remark that the choice of setting the *exponent to* 4 is carefully made. For example, setting it to 2, the accuracy for CIFAR-10 drops from $94.4\%$ to $93.1\%$, which further drops to $87.1\%$ if the absolute value is used. An exponent of 2 is much more demanding than the one of 4 on matching precisely the output energies to the rules of the coding scheme, and therefore it seems to considerably restrict the flexibility of the function of the subNNs, degrading in turn the overall performance. This trend is exacerbated with using the absolute value. We observed as well the same behavior in CIFAR-100.

### 3.1.3 DropSubNNs

The second algorithmic addition is a type of dropout [69], similar to techniques such as SpatialDropout [75], StochasticDepth [26], and DropPath [35]. Seeing each subNN as one more complicated neuron, we apply dropout to it, so its output is zeroed with a fixed probability $p_{drop}$. This method is coined as *DropSubNNs*. Our aim is to reduce the "co-adaptation" effect [69] on the subNN level, according to which subNNs collaborate in groups instead of trying to independently produce useful features. In our implementation, we apply the same random mask to all blocks that have the same coding scheme.

## 3.2 The Network

The complete network is constructed as a sequence of blocks. The Energy Normalization, $\mathcal{L}_{code,l}$, and dropSubNNs are applied only to blocks whose subNNs we want to specialize in some subsets of classes. Thus, for blocks with $r_l = N/N = 1$, we use the conventional ResNeXt block as in Fig. 1(a). In that sense, the ResNeXt model is a Coded ResNeXt model where all blocks have $r_l = N/N$.

### 3.2.1 Coding Scheme Construction

We remark that the coding scheme is constructed before training, and that the subNNs are trained to comply with this fixed, predefined, scheme. In general, the coding scheme can be arbitrary, and can possibly incorporate semantic similarities between classes. However, we aim to make a proof of concept where *it is possible to specialize subNNs to subsets of classes defined before training, even in the case when the classes within those subsets may not be semantically related*. Specifically, we found that even when the coding schemes are designed in an agnostic way with respect to the nature of the classes, good performance is guaranteed if some general construction rules are followed.

We construct one coding scheme per ratio $r_l$ so that a coding scheme is uniquely characterized by the ratio $r_l$ and any two blocks $l, l'$ with $r_l = r_{l'}$ have exactly the same coding scheme. A general rule we follow is that the deeper in the network a block is (i.e., the larger $l$ is), the smaller is the $r_l$ assigned. The first blocks have $r_l = N/N$ so that their subNNs produce low-level features, potentially useful for recognizing any of the classes. Deeper blocks have smaller $r_l$ so that their subNNs specialize on a subset of classes. [2] This rule not only is intuitive, but also works better in practice. For instance, in CIFAR-100, changing the proposed ratios (20/20, 8/20, 4/20) to (8/20, 8/20, 8/20) drops the accuracy from 78.8% to 77.9%, and inverting the order into (4/20, 8/20, 20/20) gives 76.7%.

Given a block $l$ with ratio $r_l$, we would like the coding scheme to satisfy the following three rules:

A. The number of "1"s must be equal to $N_{act,l} = r_l N$ with $N$ being the codeword length.

Moreover, we want to avoid under- or over-utilizing any subNN, in the sense of assigning too few or too many classes for it to process. As a result, the second rule is:

B. Seeing the coding scheme as a binary table, as in Fig. 1(c), the sum of each column should be approximately the same.

Finally, we aim at making the set of subNNs dedicated to work for a class, to be as different as possible from the sets assigned to the rest of the classes. This translates to:

C. The minimum Hamming distance between all pairs of codewords should be as high as possible.

Given $r_l$ and $N$, many coding schemes that follow the above rules may exist. For example, permuting the rows and/or the columns of the binary matrix in Fig. 1(c) gives new valid coding schemes. We experimentally checked (on CIFAR-10/100) that any scheme that satisfies the above properties provides similar results. Wanting to find $K$ binary codewords of length $N$ that only satisfy rule C is already an NP-Hard problem and in our case there are two additional rules. For that, we resort to an heuristic algorithm, presented in Appendix A, which finds good coding schemes according to the above rules and was used to generate the codes of all our experiments. On a high level, the algorithm first constructs the set of all binary codewords of length $N$ with $N_{act}$ ones (rule A). Second, it extracts from it a subset containing only codewords whose mutual Hamming distance is always higher than a given threshold (rule C). Finally, it extracts multiple combinations of $K$ codewords from that subset and checks which one is a good coding scheme in terms of how well rule B is satisfied. Finally, we notice that the above coding scheme has some interesting connections with constant-weight codes.

### 3.2.2 Architecture and Total Loss

We succinctly describe a Coded ResNeXt block as $[C_{out}, d, r_l]$, with $C_{out}$ being the number of channels the block outputs and $d$ being the bottleneck width as in ResNeXt [84]. A conventional ResNeXt block is expressed as $[C_{out}, d, N/N]$. Following [84], given the number of subNNs $N$, the bottleneck width $d$ is determined so that the blocks have about the same number of parameters

---

[2]In fact, the last linear layer of the ResNeXt can be seen as $K$ subNNs, each performing a simple linear combination, and the coding scheme has the lowest possible ratio $r_l = 1/K$ (i.e., codewords are one-hot vectors).

| stage | Coded ResNeXt-29 (10×11d) for CIFAR-10 | Coded ResNeXt-29 (20×6d) for CIFAR-100 | Coded ResNeXt-50 (32×4d) for ImageNet |
|---|---|---|---|
| s0 | conv $3\times3, 64$ | conv $3\times3, 64$ | conv $7\times7, 64$, str. 2, $3\times3$ max pool, str. 2 |
| s1 | $[256, 11, 10/10]\times3$ | $[256, 6, 20/20] \times 3$ | $[256, 4, 32/32]\times3$ |
| s2 | $[512, 22, \mathbf{5/10}]\times3$ | $[512, 12, \mathbf{8/20}] \times 3$ | $[512, 8, 32/32]\times4$ |
| s3 | $[1024, 44, \mathbf{3/10}]\times3$ | $[1024, 24, \mathbf{4/20}] \times 3$ | $[1024, 16, \mathbf{16/32}]\times6$ |
| s4 | global avg. pool, 10-d fc | global avg. pool, 100-d fc | $[2048, 32, \mathbf{8/32}]\times3$ |
| | | | global avg. pool,1000-d fc |

Table 1: Architecture for each dataset. A block is described by $[C_{out}, d, N_{act}/N]$, with $C_{out}$ being the number of channels it outputs and $d$ being the bottleneck width. For CIFAR architectures, stages s1, s2, s3 have approximately $0.2, 0.9, 3.5$ million parameters, respectively (in total $4.7M$). For ImageNet, s1, s2, s3 and s4 have $0.2M$, $1.2M$, $7.0M$ and $14.5M$, respectively (in total $25.0M$).

and FLOPs as the corresponding blocks of the original ResNet bottleneck architecture [22]. Table 1 presents the networks trained for CIFAR-10 (C10), CIFAR-100 (C100) [32], and ImageNet-1k (IN) [61] classification datasets. In CIFAR-10/100 we chose $N$ to be small yet sufficiently high to enable reducing $r_l$ to less than $0.25$ and still obtaining a coding scheme with minimum Hamming distance not less than 4. For ImageNet we used the default values of ResNeXt-50. Remarkably, even though the number of classes increases exponentially across datasets ($K \in \{10, 100, 1000\}$), a strong coding scheme can be found to efficiently share the subNNs between classes, so that (a) random pairs of classes are assigned to very different subsets of subNNs; and (b) only a linear increase of the number of subNNs ($N \in \{10, 20, 32\}$) is needed.

Let $\mathcal{L}_{class}$ be the conventional cross entropy loss and $B_{code}$ be the set of indices $l$ pointing to the blocks with ratio $r_l < 1$. Let $\mu$ be a loss-balancing constant; then the total loss used to train the network is

$$\mathcal{L}_{tot} = \mathcal{L}_{class} + \mu \sum_{l \in B_{code}} \mathcal{L}_{code,l}. \tag{6}$$

## 4 Experiments

In this section, we present experimental results to assess the performance of the proposed Coded ResNeXt. First, we show that our algorithm achieves subNN specialization. To demonstrate this we show that when the subNNs specialized on the class of interest are removed, the performance degrades, whereas it remains the same or even improves when the subNNs removed are not specialized for that class. To further prove the specialization, given a class, we keep only the subNNs assigned to that class. That way, we retrieve a lightweight single-purpose binary classifier, accurately deciding whether the input sample belongs to the class or not. Finally, we show that it is possible to get good predictions from intermediate blocks without evaluating the whole network. Those predictions can also be used to provide confidence on whether the final network's prediction is correct.

### 4.1 Setup and Validation Accuracy

In order to make a fair comparison with ResNeXt, on ImageNet [61] we follow the training process proposed by timm library [1]. The epochs are 250 (first 5 as warmup [20] and last 10 cooling down), the batch size is 1536, and the learning rate 0.6. RandAugment [10] of 2 layers and magnitude 7 (varied with a standard deviation of 0.5) is used and also random erasing augmentation [91] with probability 0.4 and 3 recounts. We diverge from

| | $(\mu, p_{drop})$ | Coded ResNeXt | ResNeXt |
|---|---|---|---|
| CIFAR-10 | (6, 0.1) | **94.41%** | 93.66% |
| CIFAR-100 | (6, 0.1) | **78.76%** | 76.86% |
| ImageNet | (2, 0.1) | **80.24%** | 79.50% |

Table 2: Default hyperparameters and validation accuracy. For Coded ResNeXt on ImageNet $80.24\%$ is the mean of 3 runs, which gave almost identical results ($80.21\%, 80.25\%, 80.26\%$).

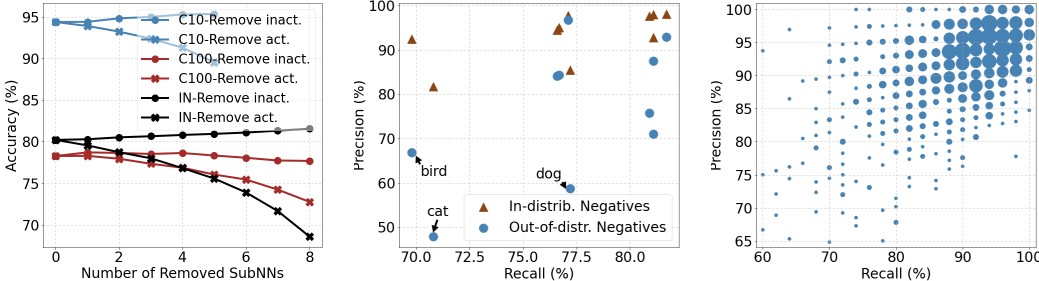

(a) Removing subNNs of a block.  (b) Precision-Recall on CIFAR-10.  (c) Precision-Recall on ImageNet.

Figure 2: Demonstrating the specialization of subNNs to their assigned set of classes. **(a)** Performance when removing active versus inactive subNNs from a specific block. **(b)** Precision-Recall from all extracted binary classifiers trained on CIFAR-10. Out-of-distribution negatives are the validation set of CIFAR-100. **(c)** Precision-Recall from all extracted binary classifiers trained on ImageNet. The larger the marker, the more points fall into that area.

the timm's proposed process only on the resolution of the input *training* images. We reduce the resolution from 224 to 160, since on the TPU-v2 of Google Colab (the platform used for our experiments) the training would take more than three weeks. Still, timm reports $79.77\%$ accuracy for the ResNeXt-50, which is clearly smaller than the one of Coded ResNeXt-50, despite being trained with lower resolution. In Appendix B we provide further details, including the training procedure for CIFAR.

In Table 2 we compare the accuracy of Coded ResNeXt against the corresponding ResNeXt (i.e., when setting all ratios $r_l$ equal to 1). We observe a clear improvement in accuracy across all datasets. Surprisingly, forcing the subNNs to specialize to specific set of classes yields significant gains even if the assignment of classes to subNNs is done in a way agnostic to the semantics of the classes. Table 2 presents the default values used for the introduced hyperparameters $(\mu, p_{drop})$ and the achieved validation accuracy. In Appendix D we perform an ablation study on those hyperparameters.

## 4.2 Specialization

A key idea of our work is to specialize each subNN to specific subset of classes; hence the first experiment is designed to test whether our architecture succeeds in achieving specialization. Assuming a subNN is assigned to activate for some class, if this subNN helps indeed on the classification process of images belonging to that class, removing this active subNN should negatively impact this process. On the other hand, if that subNN is not assigned to that class, then it should remain inactive during the process, so removing it should have no impact (degradation) on the performance.

For the first experiment we pick a block $l$ from which we randomly remove subNNs[3] in two ways. Given the class of the input image sampled from the validation set, the first way randomly removes $k \leq N_{act,l}$ subNNs from the set of active for that class subNNs. The second way randomly removes $k \leq N - N_{act,l}$ subNNs from the (complementary) set of inactive subNNs for that class. For illustration, in Fig. 2a we pick the last block of stage s2 in the architecture for CIFAR (see Table 1) and the second of stage s4 for that for ImageNet. Figures with respect to other blocks are presented in Appendix F.

In Fig. 2a, we observe the same behavior across all datasets, which confirms that the more active subNNs are removed, the more the performance degrades. Interestingly, when removing inactive ones, the accuracy tends to increase. Our interpretation is that even though the inactive subNNs are trained to output zero signal, this is never perfectly achieved in practice and their output always interferes with that of the active subNNs. Thus, taking out the interferers could improve accuracy. Note that this higher accuracy of the neural network is not actually achievable since to remove a subNN we need to know a priori the class of the input so as to know the set of (in)active subNNs for that class. Finally, we remark that even if all active subNNs are removed from one block, the

---

[3]Removing a subNN from a block in this architecture is equivalent to zeroing all of its parameters or to zeroing its output before the Energy Normalization.

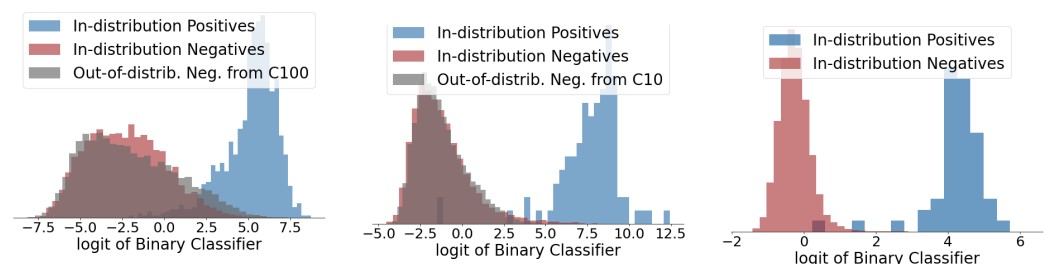

Figure 3: Output distribution of Binary Classifier (BC) of the first class (airplanes, apples, tench) of each dataset (CIFAR-10, CIFAR-100, ImageNet, respectively).

performance does not necessarily plummet. We believe that the reason behind this is that, in that case, information can still pass from the previous block to the next one through the skip connection.

### 4.3 Binary Classifier

Having confirmed that the subNNs specialize on their assigned subset of classes, we proceed with testing this property to the extreme. For that, instead of randomly removing few subNNs from one block, given a class $k$, we remove from *all* blocks all subNNs not assigned to class $k$. The rationale behind this is to check whether by keeping only the subNNs specialized on one class we can obtain a binary classifier capable of recognizing that class among the others.

In Fig. 3 we pick the first class of CIFAR-10/100 and ImageNet ("airplane", "apple", and "tench", respectively) and remove all inactive subNNs for that class. We also remove from the final linear layer everything except its first row of parameters so as to keep only the first logit corresponding to the first class. That way, we retrieve a sub-model whose output is one-dimensional. Figure 3 depicts with blue the output distribution when inputting samples of the validation set belonging to the first class of the dataset (i.e., in-distribution positives), and with red when the samples belong to some other class (i.e., in-distribution negatives). Clearly, the extracted sub-models do operate as binary classifiers giving high output when fed with samples of the class for which they are specialized. To further showcase the specialization, for the sub-models trained on CIFAR-10 (resp. CIFAR-100) we input samples that belong to the validation set of CIFAR-100 (resp. CIFAR-10). Those are considered out-of-distribution (OOD) predictions, since the sub-model has never been trained on such samples. Nevertheless, as Fig. 3 shows, the extracted BC still perform very well. A possible justification for the the good OOD performance of the extracted BC is the functional lottery ticket hypothesis [87], which states that every full network contains a subnetwork that can achieve better OOD performance.

Therefore, *with a single training of a large multi-purpose neural network, we can straightforwardly extract multiple single-purpose models that are considerably lighter (38%, 27%, and 35% of the initial parameters for CIFAR-10, CIFAR-100, and ImageNet architectures, respectively).* Given a threshold distinguishing between positive and negative predictions, each of those models becomes a BC. We set that threshold to the value maximizing the F1-score of the BC when fed with samples from the training dataset. In Figs. 2b and 2c[4] the performance of the BCs (on the validation set and the out-of-distribution set) is depicted in precision-recall plots. Notably, for CIFAR-10, the worst performance is obtained by the BC for "cats" when fed with CIFAR-100's out-of-distribution samples. This seems reasonable, since we request from the classifier to distinguish cats from classes like leopard, lion, and tiger, but without having "seen" any sample of them during training.

In ResNet, complete blocks can be removed without severely degrading the accuracy [77]. Hence, it is reasonable to ask whether the conventional ResNeXt is also robust to the removal of subNNs and thus, good BCs can be extracted from it without the need for our proposed modifications. Interestingly, this is not the case and the answer is negative. In CIFAR-10 for instance, the extracted BCs from the

---

[4]The validation set of ImageNet has 50 positives and $999*50=49950$ negatives per class. In Figure 2c we consider only $9*50=450$ randomly selected negatives to compute the precision and recall. We do that (i) in order to keep the same ratio of positives versus negatives as in CIFAR-10 and allow comparison, and (ii) because the dataset is very skewed; e.g., even a very conservative threshold that misclassifies only 1% of the negatives results into approximately 500 false positives. Since they are only 50 positives, the precision becomes 10%.

Coded ResNeXt give on average precision $93\%$ and recall $77\%$ (F1-score $F_1 = 84\%$). Attempting to extract likewise BCs from a ResNeXt leads to precision $13\%$ and recall $56\%$. Finally, given a class, the complete ResNeXt architecture can be seen as a BC by considering its output to be only the corresponding logit. Comparing such BCs to the extracted BCs seems unfair since the extracted BCs not only have $2.5$ times fewer parameters, but also have never been trained as independent models. Nonetheless, this may serve as a baseline. For CIFAR-10 this baseline gives BCs with average precision $79\%$ and recall $94\%$ ($F_1 = 86\%$). Additional details and plots are provided in Appendix F.

### 4.3.1 Why ResNeXt?

In this subsection, we provide insights on why the subNNs achieve specialization and we highlight why ResNeXt serves as the appropriate architecture upon which to build our idea. The objective of our work is to construct networks in which the per-class information is forced to flow through specific paths (determined here by the coding schemes). To achieve this, we employ (i) an operation (energy normalization) that limits how many subNNs can be activated; and (ii) a loss function forcing which ones should be activated. Intuitively, those operations should suffice for constraining the information to flow through the active subNNs. A natural question that arises is how accurate this is. Let us assume that it is accurate. Then, keeping those operations unaltered and changing only the way the subNNs' outputs are passed to the subsequent blocks should not impact the flow of information. However, if instead of aggregating them by *summation*, they are *concatenated*, the performance of the extracted BCs becomes poor (precision $< 20\%$). It seems that the concatenation inhibits the "information" to pass only through the designated paths, since the performance degrades when inactive subNNs (which in theory should not participate in those paths) are removed. Let us see why.

When concatenating the outputs of the $l$-th block, the information about which are the inactive subNNs is preserved, thus the $(l+1)$-th block may depend its operation on which subNNs of the $l$-th block provide zero output. This allows information to "leak" from the inactive subNNs. On the contrary, the information that some subNNs provide zero output is lost when adding them to the final output of the block. For that reason, the ResNeXt architecture (which aggregates the outputs by summation) is very well suited for developing our idea of controlling the information paths. Interestingly, another popular block called MBConv proposed for MobileNet-V2 [62] (and later used for EfficientNets [74]) bears a resemblance to ResNeXt block and is also a good candidate for incorporating our ideas. We elaborate more in Appendix E.

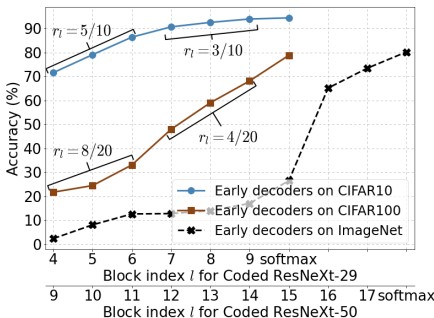

Figure 4: Accuracy of the early decoders.

|  | CIFAR-10 | CIFAR-100 | ImageNet |
|---|---|---|---|
| 0 | 52.0% | 48.7% | 46.6% |
| 1 | 67.7% | 69.5% | 70.5% |
| 2 | 77.2% | 77.2% | 87.3% |
| 3 | 83.2% | 87.7% | 90.7% |
| 4 | 89.9% | 90.7% | 90.9% |
| 5 | 92.1% | 90.6% | 91.1% |
| $\geq 6$ | 98.1% | 95.8% | 92.6% |

Table 3: Accuracy of final prediction ($\%$) given the number of early decoders giving the same prediction.

### 4.4 Early Decoding

Coded ResNeXt improves accuracy over ResNeXt while enabling the extraction of multiple lighter single-purpose models with a single training and providing transparency on how information flows throughout the network. Here we show that leveraging coding theory to design when and which subNNs should be activated allows exploiting Coded ResNeXt in other ways.

Given block $l$ with $r_l < 1$, i.e., $l \in B_{code}$, the coding scheme maps each class $k \in \{1, \cdots, K\}$ in *one-to-one* fashion to a codeword $w_{l,k}$ and then the training pushes the energies $v_l \in \mathbb{R}_{\geq 0}^N$ of the block's subNNs output to match that codeword. This allows for each $l \in B_{code}$ to measure the vector $v_l \in \mathbb{R}_{\geq 0}^N$, find the codeword $w_{l,k}, k \in \{1, \cdots, K\}$ having the minimum distance to $v_l$, and

consequently predict the class of the sample. As a result, each block $l \in B_{code}$ becomes an early decoder predicting label $\arg\min_k ||v_l - w_{l,k}||_2$, $k \in \{1, \cdots, K\}$, with $|| \cdot ||_2$ being the L2 norm. In Fig. 4 we depict the accuracy of every block $l \in B_{code}$ when functioning as an early decoder. Interestingly, as a sample passes from one block to the next one, the probability of being correctly decoded increases. In Appendix D we show that as the coefficient $\mu$ of $\mathcal{L}_{code,l}$ increases, the early decoders get improved, but past a certain point this comes at the expense of the overall accuracy.

Finally, we illustrate another possible utility of the early decoders. In Table 3 we measure the accuracy of the network's final prediction given how many early decoders also provide the same prediction. It is obvious that the more decoders agree with the final prediction, the higher the probability to be correct. Therefore, early decoders provide a confidence estimation on the correctness of the network's prediction. Specifically, they can be a source of extra features used to improve state-of-the-art confidence calibration methods [21]. We further analyze this in Appendix C.

## 5   Conclusion

In this paper, we proposed a network architecture in which the information related to each class flows through distinctive and clearly defined paths. We depart from the ResNeXt architecture and apply few yet crucial modifications that allow achieving higher accuracy and have several additional attractive properties. First, we specialize each part of the model on a specific and fixed subset of classes, which –given a certain class– enables to obtain a binary classifier for that class by keeping only the relevant parts of the network. Second, it allows to obtain early predictions without the need for entirely evaluating the network. Third, if fully evaluated, a confidence level can be produced on the correctness of the final prediction. In this work, we achieved specialization without having to rely on the semantic similarities between classes. Nonetheless, we conjecture that further gains (higher accuracy, lighter binary classifiers, smaller network, etc.) can be obtained by exploiting such similarities. Finally, while in this work we have shown that our method outperforms ResNeXt in image classification, future directions may include comparing in other computer vision tasks such as detection, few-shot learning and robustness against adversarial attacks.

## Acknowledgements

The authors are grateful to Angelos Katharopoulos and Marina Costantini for helpful discussions and remarks, and also thank the anonymous referees for their useful and constructive comments. The work of A. Avranas has been supported by the EURECOM-Huawei Chair on Advanced Wireless Networks. M. Kountouris has received funding from the European Research Council (ERC) under the European Union's Horizon 2020 research and innovation programme (Grant agreement No. 101003431).

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
