# OpenReview forum: "Towards Disentangling Information Paths with Coded ResNeXt"
_NeurIPS.cc/2022/Conference — NeurIPS 2022 Accept_

### Official Review · Reviewer_fBFb · 2022-07-11

**Rating:** 7
**Confidence:** 4
**Soundness:** 4 excellent
**Presentation:** 3 good
**Contribution:** 4 excellent

**Summary:**

The paper presents a novel way to guide information flow through a network allowing different paths to be followed for different predictions.  In the presented instantiation of the idea, the paper uses the ResNeXt network that follows different and disentangled paths for different output classes. The paper highlights the use of such a network for interpretability, as well as computational savings that can be obtained via this process. The results show that the network can achieve disentangled information paths without sacrificing any accuracy in the ImageNet-1K and CIFAR datasets.

**Questions:**

# Comments

C1: I would have loved to see a bit more analysis on how the disentanglement in the subnets translates to disentanglement (if any) in the feature space. Even a qualitative example by selecting a few distinct classes, and higher-level features that are shared/different for those classes would be a welcome addition.

# Questions

Q1. Could different coding schemes be used to allow networks to learn subnets based on other factors (e.g., metadata, additional information, hierarchical classes, etc. ). How difficult is this?

Q2: What is the effect of a larger/smaller amount of networks shared with all classes? Can you share any insights about what this "shared" portion learns?


**Limitations:**

The paper adequately addresses limitations.

**Strengths And Weaknesses:**

# Strengths

- **[S1]: Original, Interesting Idea:** The main idea in the paper is original and very interesting. Without using semantics from any classes, the proposed work can guide the information flow for different classes through different subnetworks without sacrificing overall accuracy (Though for this POC, lots of tricks need to be applied). This is a very significant innovation.

- **[S2]: Significant to a broad audience:** The paper highlights different ways where this might be useful; This reviewer agrees with all of them, and actually thinks that the significance might be broader still. A strong contender is out of distribution generalization and mitigation of spurious correlations. The paper notes that

> Interestingly, when removing inactive ones [sub-networks], the accuracy tends to increase.

I find this result to be highly encouraging towards better disentanglement, OOD generalization, and mitigation of spurious correlations. Similarly, the computational savings aspects of this network are also not sufficiently addressed here (not a weakness, just a note), but a combination of early exit possibility with well-learned subnetworks, can make this a powerful tool for extracting a single-purpose lightweight network out of a bigger network. Finally, as noted by the a paper, actually exploiting similarities/semantics/other information about the classes (which this work does not) can further improve this method in all fronts including better disentanglement, OOD generalization, accuracy etc.


# Weaknesses

- **[W1]: Some presentation-related woes:** There are some issues with clarity. Especially, I found the discussion coding schemes construction a bit hard to follow. The effects of r_l and different choices for it (including extreme cases, where a high r_l is used for most layers -- with the extreme case being the regular ResNeXt, and/or low r_l used throughout) are also not fully explored. Also missing is a bit more thorough analysis of specialization as a function of depth. However, all of these are "nice to haves" and does not really detract from the main contributions of the paper.

- **[W2]: Some breadth missing in literature review:** While the idea presented is undoubtedly novel, I do think there are similarities with adjacent areas of research, especially in out-of-distribution literature. I have listed a few below:

    - R1: Early exit networks: https://arxiv.org/abs/2004.12814
    - R2: Early exit networks used for different information processing for different samples: https://arxiv.org/abs/2204.02426
    - R3: Subnetworks used to improve OOD detection: https://arxiv.org/abs/2106.02890


# Overall

Overall, this is a very exciting work; While the scope of experiments is a bit limited (I'd have like to see more, as noted above), this is hardly an actual weakness, just something the authors/others can expand in the future. Overall, the successful demonstration of disentangled information flow for different classes is already exciting, but the potential for this idea in other applications (noted above) is also equally exciting. I would definitely like to see this at the main conference.

---

> ### Author Response · Authors · 2022-08-02
> **Response to Reviewer fBFb**
>
> We are happy to read that the reviewer finds our idea innovative, and that they share our excitement for the diverse aspects in which we believe our method is useful.
>
> **Mitigation of spurious correlations.**
> We would like to thank the reviewer for the insightful comment that removing some parts of the network (in our case inactive subNNs for a given class) may remove some spurious correlations, thus leading to a performance improvement. This seems to be a good explanation for why the extracted binary classifiers (which contain no inactive subNN) have a good performance on recognizing OOD negative samples. This is a very interesting connection to the reference R3 stating that even neural network models that  "exploit spurious correlation can contain subnetworks that capture invariant features'' and "achieve better OOD performance''.
>
> **Presentation improvement.**
> We rewrote some parts related to the construction of coding schemes to make them clearer. The ratio $r_l$ controls how specialized the subNNs of the block $l$ are. The smaller this value is, the less "1'''s  the binary matrix representing the coding scheme has and so each subNN activates for a smaller set of classes. We experimentally  verified on Cifar-10 and Cifar-100 that the deeper a block is in the architecture, the smaller its ratio should be. This means that the first blocks have subNNs that activate for all the classes and the last blocks have specialized subNNs activating only for a few classes. This is why in Cifar-100 we propose the stages (s1,s2,s3) to have respectively the ratios $(1.0, 0.4, 0.2)$. For example, when we change the decreasing values $(1.0, 0.4, 0.2)$ to constant values $(0.4, 0.4, 0.4)$ the accuracy drops from 78.8\% to 77.9\%, and inverting the order to $(0.2, 0.4, 1.0)$ gives the even worse 76.7\%. On ImageNet, where we used a deeper architecture (ResNeXt-50 instead of ResNeXt-29) we also experimented on decreasing the ratios more aggressively. Specifically, we propose for stages (s1,s2,s3,s4) the ratios $(1.0, 1.0, 0.5, 0.25)$ which if changed to $(1.0, 0.5, 0.25, 0.125)$ then the accuracy drops by more than 1\%. This suggests that pushing the subNNs to specialize too early may deteriorate performance.
>
> **Missing related literature.**
> We thank the reviewer for bringing up further relevant related work; we have now incorporated these references to better illustrate the connections of our paper to the existing literature.
>
> **Translation of subNN disentanglement to disentanglement in features \& shared network portions.**
> Unfortunately, we have not yet investigated how the disentanglement on the level of subNNs translates to the disentanglement of the features, so we hesitate to give any uncertain insights on what the subNNs that are shared across a small/big set of classes actually learn.  We do conjecture though that if codewords are designed to make the information paths of semantically similar classes to share a large portion of the network, then this portion will learn characteristic features of this "superclass" (see answer below). However, we have not yet investigated how changing these shared portions affects the learned features. These are all very interesting directions.
>
> **Using additional information to construct coding schemes**
> If additional information is available, it could be exploited for the design of coding schemes and potentially improve performance. This can certainly be done but may add some difficulties with respect to our current approach, which assigns  codes to classes/subNNs arbitrarily (see also the first reply to Reviewer 1).
> Here is an example exploiting the potentially additional information about the hierarchical structure a dataset might have:
> if each class belongs also to a superclass (e.g. "cat" and "dog" both belong to "quadruped mammals"), then each sample has both a coarse and a fine label.
> Therefore, we can design the coding schemes in a way that the first blocks have subNNs specializing in the superclasses and the last blocks have subNNs that each specialize on the classes of a specific superclass. This new coding scheme would have to follow additional requirements, since now the information paths of "cat" and "dog" depend on each other.  We expect this method to increase the interpretability of the model and to make a more efficient use of the computational power of the network thanks to "grouping features" between classes. However, we also expect that it will not allow for early decoding of the individual classes, but of the superclasses instead, i.e. it will not be able to distinguish "cat" and "dog" from the early layers' outputs, but it will confidently decode that the input is a "quadruped mammal".

---

> > ### Comment · Reviewer_fBFb · 2022-08-08
> > **Thank you for your answers**
> >
> > Thank you for your responses. I have also read other reviews and responses to them.
> >
> > - I do think the proposed presentation improvement is helpful!
> > - I really hope that there is some time and space in the paper to explore the nature of feature disentanglement and shared network portions before camera-ready but it is not a deal-breaker in terms of my scores.
> >
> > Overall, I do not believe I have read anything that means a change in my score and main points. Feel free to raise any further notes and I will do the same till the end of discussion period.

---

### Official Review · Reviewer_UmeP · 2022-07-11

**Rating:** 6
**Confidence:** 2
**Soundness:** 2 fair
**Presentation:** 2 fair
**Contribution:** 3 good

**Summary:**

The paper proposes Coded ResNext for interpretability in the classification task where each subNNs are dedicated to specific classes. Each class have distinct binary codes and subNNs are trained by the coding loss with binary codes. Each subNNs can be used as a binary classifier and early prediction is possible. In Cifar-10, 100, ImageNet, Coded ResNeXt outperforms ResNeXt.

**Questions:**

Please check weaknesses.

1) I believe the justification of ResNeXt by the authors is not enough and an ablation study is required.

2) Randomness on the binary code should be analyzed.


**Limitations:**

Yes

**Strengths And Weaknesses:**

**Strengths**
- Coded ResNeXt idea is very novel and interpretability in Coded ResNeXt was quite interesting. (I'm not the expert in this area. Therefore, I have to check the other reviews.)
- SubNetworks trained in the coded ResNext can be used as a binary classifier and early prediction is possible with the intermediate layer results.
- There are many attractive properties.

**Weaknesses**
- The main concern is that the main architecture is restricted to ResNeXt. The alation studies on the other network should be included. Even though the justification for ResNeXt is provided in section 4.3.1, I believe other multi-branch networks which aggregate the outputs by summation can substitute ResNeXt. Sub blocks for ResNet, MobileNetV2, and Efficient Net (MBConv) might substitute subNN of ResNeXt.
- The performance of this method seems to depend on the binary code. An ablation study for this randomness on the binary code for classes is required to support the superiority of this method.
- In section 3.2.1, the authors state that the coding scheme is an NP-Hard problem. However, this kind of problem can be polynomial as in [1] where the sparsity of each binary code is fixed and the pairwise term which corresponds to the hamming distance sets the binary codes apart as far as possible.  Please check section 4.2.1 and section 4.3 in [1].

[1] Efficient-end-to-end learning for quantizable networks

---

> ### Author Response · Authors · 2022-08-02
> **Response to Reviewer UmeP**
>
> We are pleased to read that the reviewer considers our idea very novel and interesting.
>
> **Other multi-branch architectures.**
> The goal of our paper was to present a method to force each branch (i.e. subNN) of a multi-branch network to specialize on a subset of classes. We chose ResNeXt because it is one of the most popular multi-branch architectures but also because  the output of all branches are conveniently aggregated by summation. In 4.3.1. we explain that this forces each block to depend only on the output of the active subNNs of the previous block, since the inactive branches are pushed to output a zero signal whose information is lost when added to the output of the active ones. As the reviewer accurately points out, our idea can be used also on top of the MBConv block used in MobileNetV2  and later in Efficient Net. Let us define the following neural network module in PyTorch (for brevity we omit the BN and ReLU function that follows after each convolutional layer) with input variables channels\_in, channels\_mid , channels\_out, $N$, $s$:\
> module = nn.Sequential(\
> nn.Conv2d(channels\_in, channels\_mid, kernel\_size=1),\
> nn.Conv2d(channels\_mid, channels\_mid, kernel\_size=3, padding=1, groups=$N$, stride=$s$),\
> nn.Conv2d(channels\_mid, channels\_out, kernel\_size=1)\
> )\
> A typical ResNeXt block is formed by simply adding a residual connection to the above module, i.e. module(x)+x where x is the input of the block. This ResNeXt block has $N$ branches/subNNs. The MBConv uses the same module (but with a ReLU6 instead of ReLU) but also forces $N=$ channels\_mid so as the second convolution to be depth-wise. Therefore, the MBConv can be seen as a type of ResNeXt block but with the number of subNNs equal to the channels\_mid and so we expect a "Coded-MobileNetV2'' to behave the same as Coded-ResNeXt. In contrast, it is not clear how our idea could be applied on top of ResNet, since it is not a multi-branch architecture. We thank the reviewer for the accurate comment and we added this discussion to the limitations section to illustrate that our idea can be also applied for MBConv.
>
> **Choice of the binary code and performance.**
> The reviewer is right on that the performance depends on the choice of the coding scheme. For example, it is better to have dense coding schemes (i.e. high $r_l$ corresponding  to a coding scheme with a lot of 1's) in the early blocks of the network and sparse (lower $r_l$) at latter blocks. As shown in Section 3.2.1, doing the opposite (i.e. assigning low $r_l$ to early blocks and making it increase as we move deeper in the network) significantly hurts performance. Having fixed the sparsity $r_l$ of a given block, there are many coding schemes that satisfy the properties described in Section 3.2.1., and thus, that are good candidates for being used in the block. For example, permutating either the rows or the columns of the binary matrix presented  in Fig 1c will give a valid coding scheme with $r_l=3/10$. We have verified in datasets Cifar-10 and Cifar-100 that those permutations do not affect performance. We explicitly added this information in Section 3.2.1.
>
> **Methods for finding the coding scheme.**
> The reviewer asks whether we could design the code scheme using the method described in Sections 4.2.1 and 4.3 in [1].
> Specifically, if in the optimization problem (5) of [1] we assume that $P$ is the identity matrix, $c_i=0$ and we rename $n_c, d, k, z_i$ respectively as $K, N, N_{act}, w_k$, we get \
> $\min_{w_1,...,w_K} \quad \sum_{i,j\neq i}{w_i^\intercal w_k}$ \
> $\textrm{s.t.} \quad  w_k\in\{0,1\}^N, \lVert w_k \rVert_1=N_{act}\quad \forall k\in\{1,\cdots,K\}$ \
> The objective $w_i^\intercal w_k$ can be shown to be equal to $2(K-Ham(w_i,w_k))$ with $Ham()$ denoting the hamming distance. Since $K$ is a constant (the number of classes), the optimization problem above maximizes the *sum* of the hamming distances between all pairs of codewords (since it minimizes $-Ham()$). In our case we want to maximize the *minimum* hamming distance between any pair of codewords in order to force clearly distinctive information paths for each class. However, maximizing the sum of hamming distances has the drawback that it allows two codewords to  be identical. Here is a small example showing this: assume we want to find $K=3$ codewords with $N=4$ binary digits out of which $N_{act}=2$ must be 1's. Maximizing the minimum hamming distance would find a solution like $w_1=1100$, $w_2=0110$, $w_3=0011$, which results in hamming distances $d_{1,2}=Ham(w_1,w_2)=2, d_{2,3}=2, d_{1,3}=4$, sum of distances $d_{sum}=8$, and minimum distance $d_{min}=2$. On the other hand, maximizing the sum of hamming distances can lead to a solution like $w_1'=1100$, $w_2'=1100$, $w_3'=0011$ which has the same  $d_{sum}'=0+4+4=8$ but $d_{min}'=d_{1,2}=0$. We now clarify in Section 3.2.1. that we want the *minimum* hamming distance to be as high as possible. We would like to thank the reviewer for the accurate remark.

---

### Official Review · Reviewer_UvD6 · 2022-07-12

**Rating:** 6
**Confidence:** 4
**Soundness:** 3 good
**Presentation:** 2 fair
**Contribution:** 3 good

**Summary:**

An algorithm is used to construct a coding scheme per-class over resblock subNNs, with a sparser coding scheme in later layers.  An auxiliary loss is used to encourage the activations to match the *fixed* coding scheme.  This specialization can then be used to construct binary classifier sub-networks from a trained model.  It also improves overall accuracy.  I think these results are interesting and on the whole the paper is well written, but the nature of the algorithm could be presented more clearly from the beginning (especially to give a flavor of what the coding scheme is).

Notes:

  -Interested in interpretable feature.  Encourage class-specific information to flow through specific paths.

  -Each path can then be used as its own lightweight binary model.

  -Many architectures have a multi-branch design, such as inception, resnext.  This is usually justified in terms of downstream performance rather than specialization.

  -Binary codeword equal with length equal to number of branches is matched to each class, indicating branches to activate per class.

  -Possible to make half-size model which works per-class.

  -Intermediate layers can make early predictions.

  -Encourage zero energy for the de-activated blocks.

  -Fixed sparsity level over the different codes (such as 3/10).  Earlier layers have less sparsity than later layers.

**Questions:**


  -Scaling to very large number of classes?

  -The way that coding is induced still feels a bit ad-hoc, for example the use of 4th power in equation 5.  Does it not work to optimize something like a binary gumbel-softmax for each code and then try to encourage only a few to be active for each class?

  -I'm still a bit confused about how the coding scheme is set per class.  Is it random?  If not, is it some kind of top-k sparse softmax with learned parameters per-class?


**Limitations:**

Limitations and ethics are well-discussed.

**Strengths And Weaknesses:**


Strengths:


  -The inactive block removal (Figure 2a) is quite nice.


  -The construction of the sparse binary classifier is also nice.


Weaknesses:


  -After checking appendix, the coding scheme seems to be a fixed and not data driven algorithm.  While the full algorithm can be in the appendix, I feel like a high-level description of how this works should be presented much earlier in the paper.


  -The writing of the paper could do a lot more to explain how it works early on.

---

> ### Author Response · Authors · 2022-08-02
> **Response to Reviewer UvD6**
>
> We thank the reviewer for their positive comments. We are pleased to know that the paper was considered interesting and well written.
>
> **Coding scheme algorithm description.**
> As the reviewer correctly remarks, in this work the coding scheme is not parameterized and so it is not learned through a training procedure; it is defined before training and it is kept fixed throughout the training. As suggested, in order to clarify the nature of the coding scheme early on, we rewrote the introduction and now we state clearly that the coding scheme is fixed and the algorithm generating that scheme is not data-driven. Moreover, in Section 3.2.1., where we mention the general properties desired for a coding scheme, we added also a high-level description of how the specific algorithm that generates that coding scheme works.
>
> **Earlier description of the method**
> According to the reviewers recommendation, we extended the introduction to provide earlier more information on how the algorithm works.
>
> **Scaling to very large number of classes.**
> The public dataset for image classification with much larger number of classes than Imagenet-1K that we know is ImageNet-22k, which has 21.841 classes. Although we would have liked to test our idea on this massive dataset,  our limited computational resources do not allow such a heavy task. However, since we show that to have a good performance, the number of subNNs required by our method increases *linearly* ($N=10, 20, 32$ for Cifar-10, Cifar-100 and ImageNet-1K, respectively) even though the number of classes increases *exponentially* ($K=10, 100, 1000$ respectively), we expect that a Coded ResNeXt with $N\approx 10\log_{10} 21.841 \approx 43$ branches will perform well for ImageNet-22k.
>
> **Alternative ways for forcing the coding scheme.**
> We find the reviewer's suggestion to optimize a binary gumbel-softmax to learn the coding scheme very interesting.
> However, we would like to draw attention to the fact that this would not guarantee the desirable properties of the coding scheme that we obtain thanks to designing it before training and then forcing it at the output of the blocks. For example, we have complete control on the information paths and make sure that each class has both its own distinctive path and the same amount of computation power assigned to it. In contrast, learning the coding scheme as the reviewer suggests would not necessarily provide the same guarantees, unless additional constraints were specified to the optimization.
> Regarding our choice of the 4th power, we simply needed a cost that would force the energies of each subNN's output to match the assigned codewords.
> Our first natural choice was using the L2 norm; however, we verified that using the 4th norm gave better results.
>
> **Assignment of codes to classes.**
> We may not have been clear, but the assignment of codewords to classes can be arbitrary. In our case, we generated the codewords and then we matched them with the classes by just following the order in which codewords and classes were listed. We made this clearer in the new version.

---

### Official Review · Reviewer_hFvq · 2022-07-13

**Rating:** 4
**Confidence:** 4
**Soundness:** 2 fair
**Presentation:** 3 good
**Contribution:** 2 fair

**Summary:**

This paper investigates multi-branch network structures from the aspect of information/feature space disentanglement, aiming to represent each class through certain path in a multi-branch network. To achieve it, the authors propose to assign classes to subNNs via a coding scheme. Output activations are calibrated by the aid of energy normalization which is regularized with a coding loss and a dropout operation. Experiments conducted on multiple datasets show the ability of the method on improving ResNeXt accuracy as well as the subNNs to the assigned class set.

**Questions:**

1. Inputs go through different subNNs in a soft-combined manner, i.e., no hard selection of certain path for each class is performed during inference. How is the performance if using hard manner instead? And how is the performance compared to assigning classes in semantic or visual correlation (which can be computed via other pretrained networks)?

2. It would be valuable to provide activation distribution of different classes on subNNs for better understanding the behavior of the method.

3. Since this method enforces the separation of features shared between classes, have the authors tested if it can improve performance on adversarial examples?

**Limitations:**

The insight behind the method needs to be further analyzed and empirically studied.  The method aims to improve multi-branch networks, which is not restricted to certain downstream cv tasks. It seems like no potential negative societal impact.

**Strengths And Weaknesses:**

Strengths:

1. The motivation of disentangling feature space of multiple classes in multiple branches is interesting and novel.  Several reasonable strategies are developed to realize the purpose, i.e., energy normalization to stabilize the flow energy in the network, coding loss to surpass the inactive subNNs, as well as dropout on subNNs to reduce feature correlation.

2. The overall framework is simple and easy to implement.

3. The paper is well written and easy to follow.

4. Some empirical analysis on the ability of subNN specialization is provided beyond classification accuracy comparison.

Weaknesses:

1. The method assigns classes via coding vectors while indeed apply a random assignment without consideration of either semantic correlation or visual similarity between classes, which is a bit counterintuitive. A family of related work investigate and leverage the class correlation to improve classification performance, e.g., [1] and [2]. The authors need to discuss the insight behind random assignment and analyze the relationship to these related works.

2. Some important ablation studies are missing in the experiments. Please refer to the ``questions”.

3. It seems contribute little to interpret representation learning in a multi-branch network. If the focus is instead the ability of pruning network to be a binary classifier, its generalization performance in downstream tasks (e.g., detection or few shot learning) is important to verify the reasonability of feature space splitting.

[1] J. Deng et al., Fast and Balanced: Efficient Label Tree Learning for Large Scale Object Recognition. In NeurIPS 2011.
[2] J. Deng et al., Large-Scale Object Classification Using Label Relation Graphs. In ECCV 2014.

---

> ### Author Response · Authors · 2022-08-02
> **Response to Reviewer hFvq**
>
>
> We thank the reviewer for the valuable suggestions. We are happy that the reviewer found the idea of our paper interesting, novel, simple to be implemented and well written.
>
> **Random vs semantic code assignment.**
> We agree with the reviewer that the assignment of set of classes to branches differs from most other proposals in the literature. For example, in [1,2] the proposed approach depends on semantic/visual similarities to assign to each node a set of classes which certainly has its merits but also there may be some limitations. On the one hand, we expect that incorporating semantic information will lead to further performance and interpretability improvements.  On the other hand, taking into account the semantic correlations adds constraints to the design of the information paths, since now the assignment should make sure that correlated classes have a large fraction of the network in common (i.e, that they share multiple  branches). Furthermore, if the sets of correlated classes have very different cardinality, special attention should be paid so that the classes of the largest sets do not end up having fractions of the network much larger than those belonging to smaller sets, potentially biasing the behavior of the network towards favoring the classification of the former.
> Another inconvenience is that, while there is yet no standard method to incorporate semantic relationships, most approaches depend heavily on external information that might be difficult to get or simply unavailable. For example, in [2] they need the WordNet dataset to import human knowledge on the hierarchical relationships between the classes of Imagenet, or they resort to a dataset like Animal With Attributes (AWA) which provides additional attributes not usually encountered in other datasets (e.g. class "zebra'' has attributes "stripes'' and "not eat fish''). In contrast, our agnostic approach that arbitrarily assigns paths to classes is simple, effective and fair between classes. We added this discussion  in Related Work and references in those papers.
>
> **Hard selection of information paths.**
> The reviewer correctly points out that the we do not make a hard selection on which subNNs the input goes through. In the ResNeXt block, all subNNs are taking the same input. In order to take hard decisions, some sort of mechanism is needed that, for any unknown sample forwarded during inference,  chooses which subNNs "have access to'' the input and which are "deactivated''.  While conducting the research for this paper we made a number of experiments in this direction. In particular, since the greatest challenge for such a method is to decide which subNNs should take the input and which should not, we tested whether we could train a small neural network placed at the beginning of each block to make this choice. Unfortunately, we could not reach a performance similar to that of the baseline with such a method.
> This is in accordance to some works  [18,38,40,42](references of the rebuttal version) where in order to improve the computational efficiency some sort of controller is used that takes hard decisions on which channels to be removed. There, even though significant speed-ups during inference are achieved, the performance of the trained network is lower than that of the same network trained without the controllers' hard decisions.
>
> **SubNN activation distribution.**
> We thank the reviewer's suggestion to add an experiment that provides the activation distribution of different classes on subNNs. In appendix F.1 we show these distributions. We see that the distribution of the output signal of the active subNNs has a bigger tail than the one of  the inactive subNNs, which clearly shows that the output of the active subNNs contains higher values. Indeed, this experiment helps to understand better how the coding loss works.
>
>
> **Additional validation.**
> The primary goal of our paper was to show that we can define (before training) arbitrary paths through which the per-class information flows and that this can allow to harness the computation power of the network for additional purposes, such as extracting binary classifiers that perform well with no additional training. We further showed that the method improves the overall performance of the network in terms of accuracy, and also, that the extracted binary classifiers are robust to out-of-distribution data.  We agree that since our proposed method aims to improve multi-branch networks, it would be interesting to consider how it performs in other tasks like detection, few-shot learning  and robustness against adversarial attacks. We have now added these to our conclusion section as good future directions.

---

### Meta-Review · Area_Chair_bqdL · 2022-08-30

**Recommendation:** Accept
**Confidence:** Certain

**Metareview:**

The authors propose a modification to ResNeXt where each sample is routed to a subset of the network. The aim is to activate only a subset of the network (subNN), in fact extract a binary classifier for each class from the trained larger network, with a significantly smaller parameter footprint. The main idea is conceptually simple -- in each ResNeXt block, pick a subset of paths for each class (e.g. 3 sub-blocks for each class) precomputed in a data-independant way via a simple binary coding scheme. To ensure that these blocks specialize to a given class a "coding loss" is added to the classic cross-entropy loss. The coding loss forces the mean energies of the subNNs (sum of squared activations, summed across C, H, W) inactive for class k to zero and those of the active subNNs to positive values. In addition, the authors add a stochastic dropout-like operation where the output of a subNN can be zeroed-out. The authors empirically validate the idea on multiple datasets and show that (1) It can improve ResNeXt accuracy, and (2) that competitive class-specific binary classifiers can be extracted post-hoc. The ablation studies clearly demonstrate that a high degree of specialization is enforced.

The reviewers appreciated the novelty and the conceptual simplicity. The method also seems practically significant, especially if it can be extended to other neural architectures. The ablation studies were well-received. After the discussion phase there are still some questions related to the rather specific choice of the 4th power for the coding loss, as well as positioning with respect to works building on the early exit ideas. Nevertheless, the reviewers agreed that this idea is novel and interesting for the larger crowd and seems practically significant. Please update the manuscript as discussed.

**Award:**

No

---

### Decision · Program_Chairs · 2022-09-14

Accept